# Oral Wine Texture Perception and Its Correlation with Instrumental Texture Features of Wine-Saliva Mixtures

**DOI:** 10.3390/foods8060190

**Published:** 2019-06-01

**Authors:** Laura Laguna, María Dolores Álvarez, Elena Simone, Maria Victoria Moreno-Arribas, Begoña Bartolomé

**Affiliations:** 1Institute of Food Science Research (CIAL), CSIC-UAM, 28049 Madrid, Spain; victoria.moreno@csic.es (M.V.M.-A.); bartolome@ifi.csic.es (B.B.); 2Institute of Agrochemistry and Food Technology (IATA), 46980 Paterna, Spain; 3Leeds, Institute of Food Science, Technology and Nutrition (ICTAN- CSIC), 28040 Madrid, Spain; laura.laguna@csic.es; 4Food Colloids and Processing Group, School of Food Science and Nutrition, University of Leeds, Leeds LS2 9JT, UK; E.Simone@leeds.ac.uk

**Keywords:** wine texture, body, viscosity, density, trained and expert panel

## Abstract

Unlike solid food, texture descriptors in liquid food are scarce, and they are frequently reduced to the term viscosity. However, in wines, apart from viscosity, terms, such as astringency, body, unctuosity and density, help describe their texture, relating the complexity and balance among their chemical components. Yet there is uncertainty about which wine components (and their combinations) cause each texture sensation and if their instrumental assessment is possible. Therefore, the aim of the present work was to study the effect of wine texture on its main components, when interacting with saliva. This was completed by using instrumental measurements of density and viscosity, and by using two types of panels (trained and expert). For that, six different model-wine formulations were prepared by adding one or multiple wine components: ethanol, mannoproteins, glycerol, and tannins to a de-alcoholised wine. All formulations were mixed with fresh human saliva (1:1), and their density and rheological properties were measured. Although there were no statistical differences, body perception was higher for samples with glycerol and/or mannoproteins, this was also correlated with density instrumental measurements (*R* = 0.971, *p* = 0.029). The viscosity of samples with tannins was the highest due to the formation of complexes between the model-wine and salivary proteins. This also provided astringency, therefore correlating viscosity and astringency feelings (*R* = 0.855, *p* = 0.030). No correlation was found between viscosity and body perception because of the overlapping of the phenolic components. Overall, the present results reveal saliva as a key factor when studying the wine texture through instrumental measurements (density and viscosity).

## 1. Introduction

Food texture is a sensory property; it is defined as “the sensory and functional manifestation of the structural, mechanical and surface properties of foods detected through the senses of vision, hearing, touch and kinaesthetics” [1]. In liquid food, the perceived texture vocabulary is rather scarce, being often reduced to the term viscosity [2]. In wines, the perceived texture is referred to as ‘mouthfeel’ [3] and arises from the changes induced by the wine to the integrity of the salivary film that coats oral mucosa. These changes are perceived by the free nerve endings (also called tactile sensors) located in the connective tissue of filiform papillae in the mouth [4]. In wines, because of the complexity conferred by its components, describing the perception of texture often involves several terms, such as astringency, body [5], unctuosity, density and viscosity [6].

Sensory and instrumental approaches could contribute knowledge about wine texture or mouthfeel. From a sensory point of view, wine tasting has an enormous tradition being generally assessed by ‘wine experts’. Wine experts include oenologists and sommeliers, among others; and their expertise allows them to award wine quality. These specialists often influence the average wine consumer as the consumers follow the experts’ awarded medals or brands, quality lists and opinion articles [7]. However, from a scientific perspective, there is no control in expert panels performance as there is uncertainty about what sensory attributes are being analysed and what their definitions are. In sensory science, for the need to control the variables as much as possible, trained panels are normally used to perform sensory descriptive analysis (QDA) [8]. QDA results are generally used to correlate sensory and instrumental measurements. Yet, for wine, finding these correlations and relating instrumental or sensory sensations to the presence or content of the principal chemical components remains a challenge. 

The wine components most cited for influencing wine texture sensations include ethanol [9], phenolic compounds [10,11,12], glycerol [13,14] and polysaccharides [15,16]. The alcohol content in wines is often between 11–14 mL/100 mL, although it can be as low as 7.5 mL/100 mL in some botrytised wines, or 15 mL/100 mL and above in some red and dessert wines. In early research, there was a suggestion that ethanol played an important role in wine body [17]. However, in later studies, it was discovered that ethanol had little or no effect on body perception or viscous mouthfeel [18,19,20,21]. Among wine polyols, the most significant is glycerol, with concentrations reaching 10 g/L in red wines and 7 g/L in white wines. From a sensory point of view, attributes associated with glycerol are oiliness, persistence and mellowness. Still, glycerol has no detectable effect in in-mouth viscosity below a concentration of 25 g/L [14]. The total phenolic content (among other composition factors) of wine depends on many factors (e.g., grape variety, growing conditions, harvest time and/or the winemaking process) [22]. Total content can vary between 40–400 mg/L in white wines, 900–1400 mg/L in young red wines and 1600–2500 mg/L in aged red wines [23]. From a sensory perspective, controversy exists regarding the size of polyphenols or phenols structure and the astringency intensity perception. Classically, it is accepted that differences between phenolic compounds produce different saliva protein precipitation. Polyphenols with an extended structure have a higher affinity to proline-rich proteins (PRPs) as the number of interaction sites increases with polyphenols size, promoting protein precipitation [24]. But previous work [10] showed that, independently of the chemical affinity and structures, when using a trained panel, the same levels of astringency was perceived, among different aqueous-polyphenol mixtures. The only difference found was that catechins were perceived as slightly bitter. Still, a later study [25] showed that the average particle size of flavanols and saliva complexes (measured by light scattering) increased with its concentration and was linked with an increment in astringency perception.

This controversy might be caused by various factors. One of them is the bitterness, felt alongside astringency and associated with polyphenols. Therefore, the difference perceived in astringency might be linked with taste perception and not with mouthfeel [26]. Another factor is the phenols lingering presence in the mouth. A recent study [27] showed that the phenolic component stayed in the mouth for more than 2 minutes. This can cause cross-mouth-contamination between samples compromising the validity of the sensory results. Finally, in wine perception, there are other macromolecular fractions, such as polysaccharides, that have been reported to influence the texture perception of wine. These components derive from cell walls of yeast (mannoproteins from *Saccharomyces cerevisiae*) [28], grapes cells (arabinogalactan-proteins) [29] and other sources (e.g., bacteria). A mixture of neutral polysaccharides (mannoproteins and arabinogalactan-protein complexes) and acidic polysaccharide (rhamnogalacturonan II) significantly increases the ‘fullness’ sensation, above that of the base wine using QDA [30].

In summary, previous scientific works have studied the relationship of ‘body’ with ethanol glycerol or polysaccharides. Still, there is no work that integrates these components in a model using human physiological conditions, such as integrating saliva, that, until now, has only been studied in terms of astringency perception. The study of wine components joined with saliva can help wineries to find instrumental measurements to adjust wine components for specific mouthfeel characteristics.

All things considered, the objective of the present study was twofold. First, to investigate instrumentally, by measuring viscosity and density, the effect of principal wine components classically related to wine mouthfeel perception (ethanol, mannoproteins, glycerol and tannins) in model based-wines (de-alcoholised) mixed with saliva. Second, to investigate the correlations of these instrumental results with human perception by using two types of panels (trained and expert). For that, artificial model-wines with one or multiple ingredients were created. It was mixed with human saliva at 37 °C, and the resulting mixture was measured for its instrumental properties (apparent viscosity and density). Then, two panels performed sensory analysis, one trained in wine mouthfeel attributes and one expert wine panel. 

## 2. Material and Methods

### 2.1. Materials

#### 2.1.1. Model-Wine Samples

A commercial white de-alcoholised wine (0.5 mL/100 mL alcohol content) was used as a base wine in all formulations (Torres Natureo Muscat 2014, Miguel Torres S.A. Winery, Barcelona, Spain). Preparation of model-wine samples (Table 1) were formulated with either the presence or absence of ethanol (E) (ethanol absolute food grade, AppliChem, Panreac, Barcelona, Spain), yeast mannoproteins (M) (Mannoplus, Agrovin, S.A. Ciudad Real, Spain), glycerol (G) (Mineral Waters, Purfleet, UK) and tannins (T) (oak tannin, Agrovin, S.A., Ciudad Real, Spain). The concentration of the different components was chosen based on their quantities in commercial red wines, except for ethanol, which was chosen to comply with a minimum legal alcohol content of wine (8 mL/100 mL). In initial experiments, a higher content of alcohol was chosen (14%), but as this was added pure, it resulted in overpowering the senses. 

All ingredients used were food grade and dissolved/dispersed in the commercial de-alcoholised wine base. Samples were prepared the day before analysis and sensory evaluation. 

#### 2.1.2. Saliva/Sample Mixtures 

Fresh human saliva (10 mL) was provided by a healthy volunteer (23 years old). Immediately after collection, the saliva was mixed with each model-wine formulation in a proportion 1:1 (*v/v*) only for instrumental measurements.

### 2.2. Methods

#### 2.2.1. Physical Measurements in Saliva/Sample Mixtures

##### Density 

A digital densitometer was used (Anton Paar density metre, Graz, Austria) at 37 °C, measuring the density of the different wine formulations and their mixtures with human saliva (see Section 2.2). Tests were performed in triplicate for each sample.

##### Flow Behaviour

Rheological tests were conducted on the mixtures of model-wine formulations, model-wine formulations with human saliva and water with saliva. The shear rate ($\dot{\gamma }$) was measured in a rotational Kinexus pro rheometer (Malvern Instruments Ltd., UK), equipped with a 40 mm cone (1°) and a plate geometry with a gap of 0.15 mm. Five hundred microlitres of each mixture were placed with a pipette onto a pre-heated plate (37 °C). Control of the temperature, to ± 0.1 °C, was by a Peltier element in the lower plate. A cover was used to maintain the temperature (37 °C) and to avoid evaporation of the samples.

Flow curves were obtained at a shear rate ($\dot{\gamma }$) ranging from 0.1 to 100 s^−1^, selected based on previous studies [31,32]. The resulting flow curves represent viscosity as a function of shear rate, and obtained results were fitted to the Ostwald de Waele model ($\eta =K{\dot{\gamma }}^{n-1}$). K (Pa s) is the consistency index, and n is the flow index. The least-square data fit was good (*R*^2^ > 0.900) in all cases. Comparisons of samples were made using apparent viscosity values at a shear rate of 100 s^−1^. Each sample was measured in triplicate.

#### 2.2.2. Wine-Oral Texture Evaluation

##### Evaluation by Trained Panel 

A panel of 8 assessors (5 women and 3 men, 20–34 years old) with one year of training in wine mouthfeel characteristics took part in this study. In a previous trial, their sensory thresholds in viscosity, astringency and alcoholic content in wine were tested [33]. 

In a preliminary session, the panellists were asked to generate the attributes perceived, relating to the mouthfeel characteristics of the six samples (Table 2). If they found some specific taste or aroma attribute key for wine discrimination, they were encouraged to record the feeling. Then, they agreed on the attributes to be chosen and their definitions (Table 2). 

In the first two sessions (around 30 minutes), the characteristics were elicited and discussed for samples W and WEMGT, as they were the most opposed of the sample set. In the following sessions, review for the commonality of all samples provided and attributes previously generated helped compilation of a final list of attributes, as shown in Table 2. The creation of their own language was under the supervision of the panel leader, who ensured that attributes were non-redundant towards samples. The aim of these sessions was for all panellists to use the same concepts allowing them to communicate precisely with each other. Finally, along with tactile (mouthfeel) attributes, two taste attributes were also included in the final attribute list (cereal taste and bitter taste).

For the formal assessment, three sessions on different days took place. Samples (20 mL) were presented monadically in separate wine glasses, each labelled with 3-digit random codes. All tests conducted were at wine serving temperature (14–17 °C). Panellists performed the test in a room isolated from odours and noise. They were advised to wait at least one minute between samples, while water, crackers and carrots were offered as palate cleansers; the six samples were tested in triplicate. The panellists were then asked to score the attributes (Table 2) at two times, during the consumption and after 10 seconds from swallowing the sample. They used a 10 cm unstructured scale (anchored from weak to strong) to score the intensity of the attributes (body, astringency, alcoholic feeling, cereal taste and bitter taste) of the samples. 

##### Evaluation by Expert Panel

Nine wine experts were recruited to take part in a blind tasting of the wines. Experts included oenology teachers, oenology students and other wine professionals (winery oenologists). The formal evaluation was made in the same way as the trained panel, including the same number of sessions, samples per session and sample order. No training sessions were performed with this panel.

### 2.3. Data Analysis 

Analysis of variance (one-way ANOVA) was applied to study the differences between the wine samples in the sensory analysis scores, density and rheology values. Tukey test was used for post hoc mean comparisons at a 95% significance level (*p* < 0.05). Paired-samples T-tests were conducted to compare the attribute scores of the two panels. 

Pearson’s correlation between the instrumental analyses results and the mean intensity scores from the sensory descriptive test by the trained panel were applied. 

All the statistical tests were done using IBM SPSS Statistics for Windows, Version 24.0. (IBM Corp., Armonk, NY, USA).

## 3. Results

### 3.1. Instrumental Measurements

#### 3.1.1. Density 

Figure 1 shows the density measurements of model-wine mixed with saliva. There were no statistical differences among samples of model-wine with saliva. However, it can be observed that the two samples that contained ethanol, WE and WEMGT, had lower density values. This was expected since the density of ethanol is lower than water (water σ at 20 °C = 1 g cm^−3^, ethanol σ at 20 °C = 0.791 g cm^−3^). Although not significant, the sample with glycerol (WG) had a higher density, followed by the sample with mannoproteins (WM) and tannins (WT). These samples with higher density also had less variability among measurements. Authors believed that the previous ingredients (glycerol, mannoproteins and tannins) helped the saliva-wine bonding stabilisation. Previous work, with the aim of understanding body perception, measured wine density [34]. It was found that density was related to the alcohol content, and in full-bodied wines, with the alcohol content and dry extract. Besides alcohol content, the perception of the body in wines has also been linked with mannoproteins [30] content and glycerol [19]. Although in previous studies, no saliva was added to wine, the results obtained in this work followed the same trend, in the way that in the presence of glycerol and mannoproteins, density increased.

#### 3.1.2. Flow Behaviour 

Although there is no agreement for the shear rate with which wine undergoes inside the mouth, the classical master curve by Shama and Sherman [31] showed that deformation of food in the mouth would be at shear rates that oscillate between 10 and 1000 s^−1^. On the other hand, as food viscosity increases, the tongue exerts higher forces; therefore, for liquids, shear rates will be lower (≈ 100 s^−1^) [32]. In the present study, the flow curve obtained was between 0.1–100 s^−1^. A shear of 100 s^−1^ was considered as the value for tongue against the palate in physiological conditions when drinking. The apparent viscosity at this value was calculated to compare the different model-wine samples with saliva (Table 3). The flow curves are presented in Figure 2.

All samples, including a control sample (water plus saliva) for comparative purposes, showed a shear thinning behaviour as apparent viscosity dropped when the shear rate increased. The sample WT was significantly the most viscous, followed by the other tannin-containing sample (WEMGT). This result agreed with a previous study [13] where instrumental viscosity increment resulted from the formation of a saliva protein-polyphenol complex. This complex has a higher hydrodynamic diameter, shown by using dynamic light scattering and negative-stain transmission electron microscopy. Formation of these complexes is via hydrogen bonding between hydroxyl groups of phenolic compounds and carbonyl/amide groups of proteins. Hydrophobic interactions between the benzoic ring of phenolic compounds and the apolar side chains of amino acids, such as leucine, lysine and proline, in salivary proteins [11,35] also contributed to this complex formation.

Samples with mannoproteins (WM) showed lower viscosity values than samples with tannins (WT). Mannoproteins are macromolecular fractions, present in wines generally used to stabilise wine flavour, colour and foam (in sparkling wines) [36]. In this case, it is believed that the viscosity increase is due to the size of the molecules in the wine mixture. When mannoproteins and tannins were present in the same wine (WEMGT), the viscosity was slightly lower because of their influence on the tannins size [37], as mannoproteins act by encapsulating polyphenols, thus interfering in the protein binding [38]. 

The least viscous samples were WE and WG, with no significant differences among them. Previous authors, using an increasing quantity of ethanol and glycerol, investigated their relationship with the contribution of ‘body’ in wine [39]. Authors found that in ranges of ethanol content 0–15% (*v/v*) and glycerol 0–20 g/L, the viscosity varied linearly; for every 1% increase in ethanol concentration, viscosity increased by 0.047 × 10^−3^ mPa s; for every g/L increase in glycerol concentration, viscosity increased by 0.005 × 10^−3^ mPa s. However, authors only measured the wine samples without the addition of saliva, and not considering its interaction. Further to this, in this present work, the ranges of ethanol and glycerol added were smaller. Therefore, the addition of saliva and its diluting effect, plus the quantity of glycerol and ethanol added, had led to non-significant changes to WE and WG samples.

### 3.2. Descriptive Sensory Evaluation

#### 3.2.1. Wine-Oral Sensations

Table 4 shows the mean scores of sensory attributes in the mouth (4a) and after swallowing (4b) for both trained and expert panels. Descriptive sensory techniques can be used when searching for sensory-instrumental relationships [8]; therefore, first, results from the trained panel have been analysed.

Significant differences among the model-wines were found between their body and intensity perception (*p* < 0.05). Average body perception ranged from 2.71 (W: control) to 4.57 (WG: wine with glycerol), as shown in Table 4. Model-wine containing mannoproteins (WM) and model-wine containing mannoproteins and glycerol (WEMGT) were also scored with high intensity, with no significant difference from WG. 

Both components (mannoproteins and glycerol) have been previously reported as influencers in wine mouthfeel [18,19]. Mannoproteins are composed of polypeptides and linked to highly branched mannose side chain by glycosidic bonds. This big molecule (800,000 Daltons) has been previously linked with a ‘fullness’ sensation when using a trained panel [30]. Present authors believe that because of the size of this molecule, the wine acquires a structure that is perceived as bulkier mouth feeling. Therefore, the trained panel considered this wine sample as with more body. 

In the case of glycerol, the previous bibliography has not linked it specifically to ‘body’. It has been linked with other various attributes, such as oiliness, persistence and mellowness [14]. In a tribological study, it was found to decrease the friction coefficient [13]. However, the definition of the body itself is still a pending subject. In fact, there is no consensus in the reference material to be used and can include dairy products, such as whipping cream [40], gels [19] or xanthan gum [41]. 

Astringency perceived was higher for samples with added oak tannin (T). The added tannins formed complexes with the salivary protein; these complex formations have been previously described by two different bond types: hydrogen bonding (between hydroxyl groups of phenolic compounds and carbonyl/amide group of protein) and by hydrophobic bonding (between the benzoic ring of phenolic compounds and the apolar side chains of the salivary protein’s amino acids) [35]. This leads to the formation of a complex caused by proteins in saliva precipitating. Therefore, the salivary film that covers the oral mucosa loses its structure and separates from the oral mucosa causing the mouth to become dry [24]. When mannoproteins and ethanol are present, a saliva-polyphenol complex formation explains their interference and a decrease in the perception of astringency [35,36]. Other samples with the same quantity of tannins showed less astringency. In a previous study, authors found that hydrodynamic diameter of saliva with tannins increased (by a factor of almost 2.5–3); but in the presence of ethanol or glycerol, it decreased [13]. This might be due to the ability of ethanol to interfere with wine polyphenol-PRPs interactions [42]. Further, mannoproteins interferes in the tannin aggregation [37] by three mechanisms: i) encapsulating polyphenols and interfering with their ability to bind proteins [38], ii) enhancing polyphenols solubility in an aqueous medium through the form of protein-polyphenol aggregates [38], and iii) by binding salivary proteins, avoiding the polyphenols attachment [26].

In addition, as it can be seen in Table 4, for the trained panel, other samples not containing tannins also contributed to the astringency feeling. This is because, in wine, astringency is also produced by the low pH of the samples [43,44,45] due to salivary protein precipitation.

Ethanol perception was inevitably higher for the sample containing ethanol (WE). Ethanol is an effective sensory stimulant, activating brain gustatory circuits, as well as trigeminal pathways, sensitive to an irritant stimulus [46], providing sensations of hotness and bitterness that linger in mouth [47,48]. Besides alcoholic feeling, ethanol presence also produced a significant bitterness perception. In a more complex wine-model matrix, the alcohol feeling was shown to be decreased (WEMGT), and this was caused by its interaction with tannins [49]. Another reason could be the sensory perception displacement effect when presenting a different stimulus at the same time.

Besides its initial sensory profile with the trained panel, the attribute means were analysed for significant differences (*p* < 0.05) between panels (paired comparison) for each sample, and between samples within each panel (Tukey’s test). There were significant differences (*p* < 0.05) in the perception of ‘body’ between panels for the samples containing ethanol. The expert panel found ethanol to be the most influential component in body perception (WE = 4.96 ± 1.63, WEMGT = 4.77 ± 1.36) whilst the trained panel perceived ‘body’ significantly higher for samples containing glycerol, mannoproteins or both (WG = 4.56 ± 1.5, WM = 4.09 ± 1.5, WEMGT = 4.01 ± 2.1). Previous work [28] agrees with the results of both panels, as all of three components, glycerol, mannoproteins and ethanol, are influencers on the wine body [28].

For ‘astringency’, there were significant differences (*p* < 0.05) between panels, as the trained panel scored the astringency significantly higher (Table 4a). Both panels found the sample WT significantly more astringent, followed by WEMGT. Therefore, here, tannins led to astringency perception, but in the presence of other components (ethanol, mannoproteins and glycerol), the intensity of the astringency was significantly lower. 

Both trained and expert panels associated ethanol with the perception of ‘alcohol feeling’, with no significant difference between their scores. As expected, the sample with higher ‘alcohol feeling’ was WE, containing only the wine and ethanol; followed with a lower score, the sample that contains ethanol plus other ingredients (WEMGT).

The attribute ‘cereal taste’ scored significantly different between panels for all samples (Table 4a). The trained panel could not differentiate the perception of the ‘cereal taste’ compared to the other attributes. Whilst the expert panel discriminated the sample according to this attribute, scoring it higher in those samples containing tannins. This finding that oak tannin leads to ‘cereal taste’ has never been reported. The present authors hypothesised that the ‘bitterness’ provided by oak tannins could lead panellists to associate it to the ‘cereals taste’.

For ‘bitterness’, both panels agreed that samples containing ethanol and/or tannins are significantly more bitter than the rest (Table 4a), which is in relation to the ‘cereal taste’ reported previously.

#### 3.2.2. After-Swallowing Sensations

Table 4b shows the after-swallowing sensation scores when consuming the samples. The intensity of all attributes decreased, especially those regarding flavour (‘cereal’) and taste (‘bitterness’), whilst ‘body’ and ‘astringency’ values remained similar. There were no significant differences between samples within panels.

It is noteworthy that the attribute ‘body’ was detectable after swallowing the wine, showing that it is a sensation related, among others, to lingering feelings in-mouth. According to a previous study using tribological techniques [13], samples with glycerol showed higher lubrication. This would indicate that ‘body’ relates to ‘unctuosity’. ‘Astringency’ is a sensation that also lingers in the mouth, this was not unexpected for the depletion of the mouth’s mucous layer caused by polyphenols components [50] and the consequent ‘dry’ feeling.

### 3.3. Relationships between Instrumental Measurements and Sensory Analysis 

To investigate the relationship between instrumental and sensory measurements, trained panel scores and instrumental correlation analyses were performed (see Table 5). The scores of the trained panel were selected because they discriminated more between samples compared to those of the expert panel. In the same way, as the analysis was used for in-mouth scores, they showed higher values for all attributes’ intensity.

Figure 3 shows the relationship between density and ‘body’ perception for model-wine samples, with and without saliva. In the absence of saliva (Figure 3a), there was no linear relationship between instrumental and sensory ‘body’ assessment. For measurements made on samples with added saliva (Figure 3b), which mimic a real consumption scenario, a trend between density and ‘body’ appeared. Considering only the samples not containing ethanol, there was a positive and significant correlation between instrumental density and ‘body’ perception (*R* = 0.971, *p* = 0.029).

A past study demonstrated that for Newtonian fluids, the sensory attribute ‘thick’ had a high correlation with instrumental viscosity [51]. However, there are no references for non-Newtonian fluids. So, for wine-model samples with saliva, there was a small, non-significant relationship between ‘body’ and instrumental apparent viscosity in samples with added saliva, at a shear rate of 100 s^−1^.

Apparent viscosity values were correlated to ‘astringency’ values (*R* = 0.855, *p* = 0.030) (Figure 3c), confirming that the formation of the protein-polyphenol complex influenced the instrumental viscosity.

Previous work had stated that wine body classification (full bodied, medium bodied and light bodied) was empirical [18,19,20,21]. But in this present paper, authors present that the sensory wine attribute ‘body’ relates to the instrumental density, although not with viscosity.

## 4. Conclusions

The present study shows that glycerol and/or mannoproteins contributed to the ‘body’ feeling, that was correlated with the instrumental measurement of the density of model-wine samples mixed with saliva. As reported previously, tannins were the leading cause for the ‘astringency’ feeling and an increase in instrumental viscosity due to the formation of salivary protein-polyphenols complexes. In parallel, this study shows that there is a gap between the abilities of the expert and trained panels for describing wine texture sensations. This makes it even more difficult to understand the expectations and preferences of real consumers regarding wine attributes, such as ‘body’, which is normally considered the driver for wine quality perception and liking.

Future studies will include real wine samples and saliva for multiple participants. There is a need for further research on how consumers understand the term ‘wine body’ in commercial samples. Also, determining what are the characteristics of the wine consumers associate with the term ‘body’ will be beneficial. In addition, knowing how the composition of different wines affects the values of physical properties, such as density or viscosity, will be of immense value.

## Figures and Tables

**Figure 1 foods-08-00190-f001:**
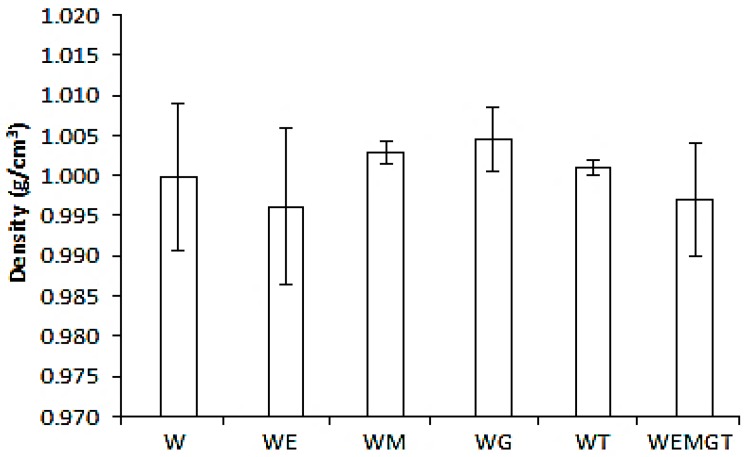
Density measurements of model-wine formulations with saliva. Bars represent the standard deviation. Model-wine letters indicate components: base wine (W), ethanol (E), mannoprotein (M), glycerol (G) and tannins (T).

**Figure 2 foods-08-00190-f002:**
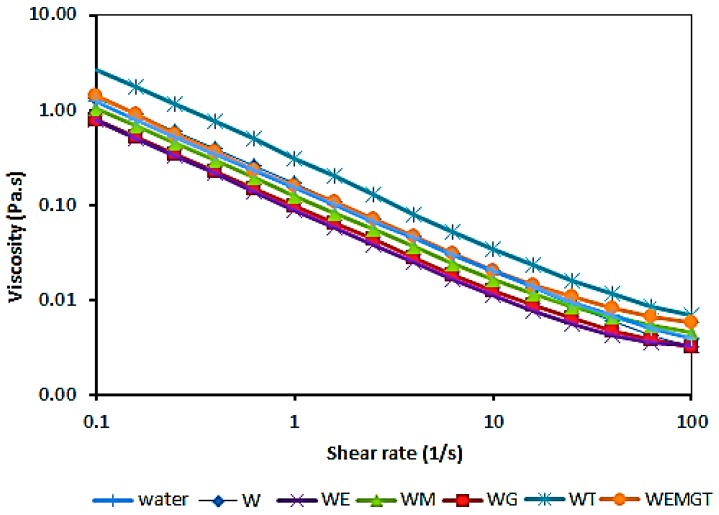
Dynamic viscosity of model-wine formulations with human saliva (1:1). W (♦), WG (■), WM (▲), WEMGT (●), WE (X), WT (X), water (l). Model-wine letters indicate components: base wine (W), ethanol (E), mannoprotein (M), glycerol (G) and tannins (T).

**Figure 3 foods-08-00190-f003:**
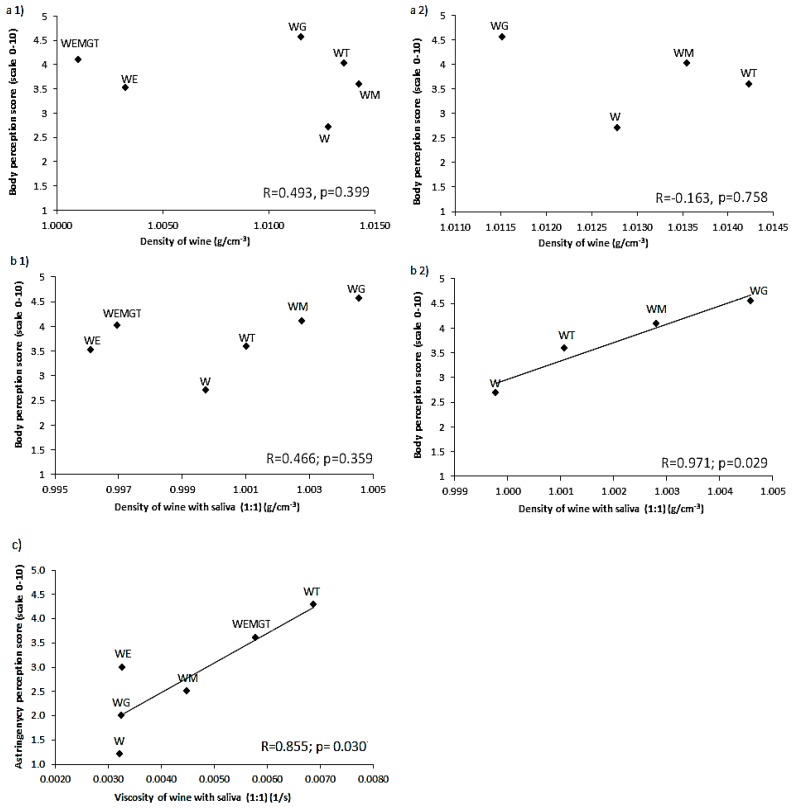
Relationship between instrumental measurement of the model-wine formulations and trained panel scores for: (**a1**) density of samples without saliva, and ‘body’ scores; (**a2**) density of samples without added ethanol plus saliva, and ‘body’ scores; (**b1**) instrumental density of samples plus saliva, and ‘body’ score; (**b2**) instrumental density of samples without ethanol plus saliva, and ‘body’ score; (**c**) instrumental viscosity of samples plus saliva and ‘astringency score’. Model-wine letters indicate components: base wine (W), ethanol (E), mannoprotein (M), glycerol (G) and tannins (T).

**Table 1 foods-08-00190-t001:** Model-wine samples made of the base wine (W), ethanol (E), mannoprotein (M), glycerol (G) and tannins (T).

Sample	Base Wine (mL)	Ethanol (mL)	Mannoproteins (g)	Glycerol (g)	Tannins (g)
W	100	-	-	-	-
WE	92	8	-	-	-
WM	100	-	0.5	-	-
WG	100	-	-	1	-
WT	100	-	-	-	0.1
WEMGT	92	8	0.5	1	0.1

**Table 2 foods-08-00190-t002:** Sensory attributes selected by the trained panel and its consensus definitions.

Term	Definition
Body	Viscosity sensation when swishing
Astringency	Dryness from the tongue tip to the throat
Alcoholic feeling	Hot sensation typical in alcoholic beverages
Cereal taste	Feeling of cereal taste
Bitter taste	Bitter taste at the end of the tongue

**Table 3 foods-08-00190-t003:** Apparent viscosity (Pa s) of model-wines at a shear rate of 100 s^−1^.

Model-Wine Samples	Ŋ (at $\dot{\gamma }$ 100 s^−1^)
W	0.0167 ± 0.002 ^ab^
WE	0.0110 ± 0.001 ^a^
WM	0.0200 ± 0.002 ^ab^
WG	0.0117 ± 0.002 ^ab^
WT	0.0347 ± 0.002 ^c^
WEMGT	0.0203 ± 0.002 ^b^

Same letters indicate no significant differences among samples (*p* < 0.05). Model-wine letters indicate components: base wine (W), ethanol (E), mannoprotein (M), glycerol (G) and tannins (T).

**Table 4 foods-08-00190-t004:** Mean of descriptive sensory scores in the mouth for trained panel and expert panel for wine-in-mouth sensations when (**a**) consuming the samples and (**b**) 10 seconds after swallowing.

	Trained Panel					Expert Panel				
	Body	Astringency	Alcohol	Cereal	Bitter	Body	Astringency	Alcohol	Cereal	Bitter
(a)										
W	2.71 ^b^*	1.40 ^b^	2.84 ^bc^	0.52 ^a^	0.56 ^b^	3.62 ^b^	2.04 ^b^	1.07	2.46 ^c^	1.71 ^c^
WE	3.53 ^ab^	2.61 ^ab^	4.99 ^a^	0.96 ^a^	4.07 ^a^	4.96 ^a^	3.73 ^ab^	4.56	2.59 ^c^	4.30 ^a^
WM	4.09 ^a^	2.92 ^ab^	2.36 ^c^	1.25 ^a^	1.42 ^b^	4.18 ^ab^	2.73 ^ab^	1.46	3.06 ^bc^	1.91 ^c^
WG	4.57 ^a^	3.50 ^ab^	2.08 ^c^	0.81 ^a^	1.18 ^b^	4.33 ^ab^	2.88 ^ab^	2.31	2.298 ^c^	1.69 ^c^
WT	3.60 ^ab^	5.28 ^a^	2.37 ^c^	1.65 ^a^	2.15 ^ab^	3.92 ^ab^	4.28 ^a^	2.56	4.21 ^ab^	2.84 ^b^
WEMGT	4.02 ^a^	4.369 ^ab^	3.54 ^b^	1.71 ^a^	3.89 ^a^	4.77 ^a^	3.88 ^ab^	4.03	4.52 ^a^	3.43 ^b^
(b)										
W	1.34 ^a^	1.33 ^a^	0.34 ^a^	0.67 ^a^	0.68 ^a^	3.50 ^a^	3.00 ^a^	1.67 ^a^	2.00 ^a^	1.12 ^a^
WE	2.01 ^a^	3.00 ^a^	3.43 ^a^	1.00 ^a^	2.33 ^a^	4.00 ^a^	3.70 ^a^	4.00 ^a^	2.33 ^a^	3.33 ^a^
WM	2.00 ^a^	3.08 ^a^	0.66 ^a^	1.57 ^a^	1.00 ^a^	3.60 ^a^	2.67 ^a^	2.00 ^a^	2.67 ^a^	1.67 ^a^
WG	2.05 ^a^	1.98 ^a^	1.00 ^a^	1.65 ^a^	1.15 ^a^	3.23 ^a^	2.33 ^a^	1.70 ^a^	1.80 ^a^	1.67 ^a^
WT	2.37 ^a^	4.02 ^a^	0.69 ^a^	1.00 ^a^	2.00 ^a^	3.31 ^a^	4.00 ^a^	2.20 ^a^	3.50 ^a^	1.95 ^a^
WEMGT	2.67 ^a^	3.68 ^a^	2.43 ^a^	2.00 ^a^	3.00 ^a^	3.98 ^a^	3.68 ^a^	3.00 ^a^	4.00 ^a^	2.67 ^a^

Model-wine letters indicate components: base wine (W), ethanol (E), mannoprotein (M), glycerol (G) and tannins (T). * Tukey test among wine-model formulations, same letter in the same column, does not differ significantly, *p* < 0.05.

**Table 5 foods-08-00190-t005:** Pearson’s Correlation Coefficient between viscosity at different shear rates and sensory attributes (body perception and astringency).

Shear rate (1 s^−1^) (Instrumentally Measured)	Body (*p*-Value)	Astringency (*p*-Value)
**100**	0.485 (0.329)	**0.855 (0.030)**
10	0.083 (0.876)	0.582 (0.226)
1	0.032 (0.953)	0.553 (0.255)
0.1	0.118 (0.823)	0.616 (0.192)
*K*	0.077 (0.884)	0.603 (0.205)
*n*	0.128 (0.809)	0.692 (0.127)

Values in bold are different from 0 with a significance level alpha = 0.05. Model-wine letters indicate components: base wine (W), ethanol (E), mannoprotein (M), glycerol (G) and tannins (T).

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
