# Peer review of "Oral Wine Texture Perception and Its Correlation with Instrumental Texture Features of Wine-Saliva Mixtures"

_foods, 2019, doi:10.3390/foods8060190_

Reviewer 1 Report

The manuscript "Oral wine texture perception and its correlation with instrumental texture features of wine-saliva mixtures"by Laura Laguna and colleaguesis of potential importance to increase the knowledge about the human perception of wine texture by the use of model wine solutions and also their relationship with the instrumental determination. The manuscript is important and contribute to understand important oral sensations which occur during wine tasting. Thus, in my opinion, the topic described is relevant and of interest for the readers, particularly for wine sector.

Thus, after a careful analysis of the manuscript, I have only a few specific comments/suggestions.

Line 40: Change “... terms astringency, body ...” for ““..., such as, terms astringency, body ...”.

Line 41: Change “.. contribute knowledge to wine ...” for “... contribute to the knowledge about the wine ..”.

Line 43: “include oenologist, ....” add also “winemakers”.

Line 65-67: I have some debuts about this sentence. Is it true ? astringency has the same intensity, independently of phenols structure and polyphenols size ? In my opinion, the degree of polymerization of some phenolic compounds, such as, proanthocyanidins, could determine also the red wine astringency.

Line 77: “... in model wines mixed ...” It is not totally correct, if these model wine solutions have no alcohol.

Line 87: Authors used “oak tannins”. Why the reason for used commercial “oak tannins” ? and not commercial tannins extracted from grape skin or seeds. In my opinion, the use of tannins from the skins or seeds will be more usual than the application of oak tannins, where it is intended to introduce some “woody character” to the wines.

Line 173: However, it is not show in the figure the statistical analysis made (analysis of variance). Introduce any letters for a better identification of statistical differences or not. In addition, in figure 1, some samples showed high standard deviation. Any tentatively explanation.

Line 192: “(WEMGT)” I’m not sure, but this sample code it is not well explained in model wine solutions explanation of table 1. Revise it.

Line 306-323: In my opinion, introduce in conclusions section only the key-points of the results from the research work.

Author Response

The manuscript "Oral wine texture perception and its correlation with instrumental texture features of wine-saliva mixtures" by Laura Laguna and colleagues is of potential importance to increase the knowledge about the human perception of wine texture by the use of model wine solutions and also their relationship with the instrumental determination. The manuscript is important and contribute to understand important oral sensations which occur during wine tasting. Thus, in my opinion, the topic described is relevant and of interest for the readers, particularly for wine sector.

We appreciate all the encouraging comments from Review 1, as well as all the suggestions to improve the paper.

Thus, after a careful analysis of the manuscript, I have only a few specific comments/suggestions.

Line 40: Change “... terms astringency, body ...” for ““..., such as, terms astringency, body ...”.

Thank you, it has been changed.

Line 41: Change “... contribute knowledge to wine ...” for “... contribute to the knowledge about the wine ...”.

Thank you, it has been changed

Line 43: “include oenologist, ....” add also “winemakers”.

Thank you, it has been changed

Line 65-67: I have some debuts about this sentence. Is it true? astringency has the same intensity, independently of phenols structure and polyphenols size? In my opinion, the degree of polymerization of some phenolic compounds, such as, proanthocyanidins, could determine also the red wine astringency.

Reviewer is right, as it is worded, it leads to a false statement and authors believe there is still controversy on this point but authors have rewritten this statement, please, see lines 78-90.

It is generally accepted that differences between phenolic compounds structure produce different affinity towards saliva in such a way that polyphenols with extended structure have higher affinity to PRPs, smaller polyphenols can bind with one phenolic ring, whilst larger polyphenols interact in a multi dentate fashion, occupying two or three consecutive prolines. In other words, the number of interaction sites increases with the size, this will help in the protein precipitation (Cala 2012).

However, the work of Ferrer-Gallego (2014) showed that despite the chemical affinity, using a sensory trained panel to evaluate the astringency of different aqueous-polyphenols mixtures, with different structures (phenolic acids: coumaric, caffeic, gallic and protocatechuic acids, and catechins: epicatechin and catechin), the same levels of astringency were perceived; the only difference was that catechins were perceived as slightly bitter. In a latter study (2016), the same authors, by measuring dynamic light scattering, found that the average particle size of flavanols and saliva complexes increases with its concentration, also showing an increment of astringency perception Ferrer-Gallego et al. (2016), as numerous previous studies had also showed.

We believe that it is important to bear in mind that the feeling of astringency is always felt alongside bitterness, so, in some of the cases where panellists were evaluating phenolic compounds at different concentration, the difference in intensity that they may perceive could be linked with taste and not with mouthfeel.

Line 77: “... in model wines mixed ...” It is not totally correct, if these model wine solutions have no alcohol.

Reviewer is right, we have changed to in model base wines (dealcoholised wine), see Line 117

Line 87: Authors used “oak tannins”. Why the reason for used commercial “oak tannins”? and not commercial tannins extracted from grape skin or seeds. In my opinion, the use of tannins from the skins or seeds will be more usual than the application of oak tannins, where it is intended to introduce some “woody character” to the wines.

That is a very good point made by the reviewer. In a preliminary study, we tried also commercial gallic acid, but sensory wise, the oak tannin was more potent and easier to identify for the trained panel.  On the other hand, oak tannins are also commercialized by different oenological companies, in order to accentuate the sensations of body and volume in the mouthfeel (http://www.agrovin.com/en/categoria-producto/tannins/?filters=product_tag[843]).

Line 173: However, it is not show in the figure the statistical analysis made (analysis of variance). Introduce any letters for a better identification of statistical differences or not. In addition, in figure 1, some samples showed high standard deviation. Any tentatively explanation.

Reviewer is right, authors did not put any letter as they were not significantly differences (see line 218-219)

Regarding the bars size, this lack of significant difference was probably due to that once the samples are mixed with saliva, the dilution effect is higher than the subtle differences because of the addition of the different wine components.

Line 192: “(WEMGT)” I’m not sure, but this sample code it is not well explained in model wine solutions explanation of table 1. Revise it.

Thank you for noticing, it is clearly a mistake and authors has been corrected

Line 306-323: In my opinion, introduce in conclusions section only the key-points of the results from the research work.

Following reviewer suggestion, we have left the facts of this study and the future lines (suggested by reviewer 2)

Reviewer 2 Report

The manuscript is well written and certainly of interest to the field. Minor corrections are suggested prior to publication as detailed below:

Table 1: correct the final sample name (should be WEMGT)

Line 105: Include volumes of saliva recovered for analysis.

Line 134: Use the letters for the sample name in the same order as shown in Table 1 (e.g. WEMGT) for consistency.

Results and discussion: Suggest that this section is redrafted to lead with the most interesting/ important results

Line 177: Include references for previous related results or expectations from the literature or highlight that this is a novel finding.

Table 3 and Figure 2: Include full sample names in the footnotes or captions for clarity.

Lines 207-210: Provide more discussion of these results. Discuss what each variable does to model wine viscosity, then compare the effects of each additive and of the combined sample. Then compare findings with expectations based on current literature.

Line 207: add (WT) after ‘tannins’ for clarity.

Line 228-251: For each sensory attribute, present the findings of the trained panel to show how each additive relates to each attribute. Then compare to the results of the expert panel. The results of the trained panel are of greater interest to the reader at this point although it is worth noting the differences between panels.

Line 242: This is an interesting result and warrants further discussion. It shows that other wine components reduce the perception of higher alcohol. Compare this outcome with expectations from the literature.

Table 4: Specify if it is the same letter within a column that indicates the result is not significantly different. Add the letters for Alcohol in 4a expert panel.

Lines 281-283: Provide more discussion about the non-significant results and compare to expectations.

Line 317-319: Remove reference to the study’s “limitations”. The study forms a good preliminary investigation into the relationship between sensory and instrumentally measured textural parameters and therefore achieved its aims. Future research is certainly warranted based on these results. Change this sentence to emphasise the needs of a future project. E.g. “Future studies will include real wine samples and saliva from multiple participants. We also need to understand how consumers understand the term ‘wine body’…”

Author Response

Reviewer 2

The manuscript is well written and certainly of interest to the field. Minor corrections are suggested prior to publication as detailed below:

Thank you very much

Table 1: correct the final sample name (should be WEMGT)

Thanks for noticing, it has been corrected

Line 105: Include volumes of saliva recovered for analysis.

Thank you, it has been included

Line 134: Use the letters for the sample name in the same order as shown in Table 1 (e.g. WEMGT) for consistency.

Thanks for noticing, it has been corrected

Results and discussion: Suggest that this section is redrafted to lead with the most interesting/ important results

Line 177: Include references for previous related results or expectations from the literature or highlight that this is a novel finding.

Following reviewer’s suggestion more bibliography has been added (Please see Line 223 to 231)

Table 3 and Figure 2: Include full sample names in the footnotes or captions for clarity.

Thank you for the suggestion, it has been included

Lines 207-210: Provide more discussion of these results. Discuss what each variable does to model wine viscosity, then compare the effects of each additive and of the combined sample. Then compare findings with expectations based on current literature.

Thank you for the suggestion, this section has been rewritten (see Lines 263-281)

Line 207: add (WT) after ‘tannins’ for clarity.

Thank you, it has been added

Line 228-251: For each sensory attribute, present the findings of the trained panel to show how each additive relates to each attribute. Then compare to the results of the expert panel. The results of the trained panel are of greater interest to the reader at this point although it is worth noting the differences between panels.

Thank you for the suggestion, this section has been rewritten and more bibliography has been also added (see Lines 291-348)

Line 242: This is an interesting result and warrants further discussion. It shows that other wine components reduce the perception of higher alcohol. Compare this outcome with expectations from the literature.

Following reviewer’s suggestion, more bibliography has been added

Table 4: Specify if it is the same letter within a column that indicates the result is not significantly different. Add the letters for Alcohol in 4a expert panel.

Thank you, it is now specified that letters in the same column does not differ significantly

Alcohol scores for the trained panel did not differ across samples, therefore, it has the same letters

Line 317-319: Remove reference to the study’s “limitations”. The study forms a good preliminary investigation into the relationship between sensory and instrumentally measured textural parameters and therefore achieved its aims. Future research is certainly warranted based on these results. Change this sentence to emphasise the needs of a future project. E.g. “Future studies will include real wine samples and saliva from multiple participants. We also need to understand how consumers understand the term ‘wine body’…”

Following reviewers’ advice, limitations have been removed and sentences have been reworded.

Reviewer 3 Report

This paper addresses a topic of interest aimed at investigating wine texture knowledge using model-wines and the correlation of instrumental texture with human perception using trained and expert panels. As I will explain, it contains conceptual and structural weaknesses, which require a major revision of the work.

Lines 8. In the abstract, the presentation of the research objectives is not clear. The present formulation does not appear effective.

Lines 32 and the followings. The introduction does not highlight the background (i.e. context) in which the research is inserted, which must, therefore, be developed into the contents:

• What is the relevance of this paper in regard to the objectives? What is the relevance for the operators of the wine sector and/or research?

• Which are the components of the study’s originality in relation to the knowledge of the aspects highlighted by the research?

• In the final part, it is also pointed out that the structure on which the paper is arranged is not presented.

Lines 81-156. The methodological part should be reviewed and better structured:

• Material and Methods should be used as a title and Model-wine samples as subtitles (line 81).

• There are paragraphs with a few lines and excessive spacing. In the current format, they look quite poor as self-standing sections.

• It should be both better articulated in the structure and connected to the objectives of the research.

Line 100. In the current format the Table 1 header is not well presented:

• The heading of the titles is not appropriately formatted;

• Mannoproteins lack of measurements indications used;

Line 104. Why was a 1/1 dosage chosen in the proportion of saliva addition to wine models? Why have solutions not been prepared with different levels of saliva added to wine models?

• An explanation should be provided to support the choices made in the implementation of the experimental design concerning the objectives of the study.

Line 150. The defining aspects reported in Table 2 do not appear well conceptualized and should be better specified:

• For example, they recall the term to be defined;

• Concerning the terminology used for the term “body” I do not agree on the definition provided because body is not only defined by the visual aspects, but the taste aspects should also be taken into consideration.

• It is also advisable to add specific references;

Lines before 305. A separation between discussions and results can help readers to understand the article.

The conclusions should be better linked to the research objectives and the conceptual framework used. We also recommend:

• To better formulate the statement on lines 308-309 which does not appear clear.

• To comment on the difficulties of knowledge in the limitations of the study, through instrumental and sensorial analysis linked to a complex food such as wine, where the interaction between its multiple components is yet the subject of open discussion among researchers.

Even the English language style still needs improvements, to make it more readable. Below one can find some examples:

·         Lines 18 to 20. The sentence has no clear meaning.

·         Lines 75 to 78: Once again the sentence has a very unclear meaning

·         Lines 182 to 189. The words shear rate were repeated 5 times.

Author Response

This paper addresses a topic of interest aimed at investigating wine texture knowledge using model-wines and the correlation of instrumental texture with human perception using trained and expert panels. As I will explain, it contains conceptual and structural weaknesses, which require a major revision of the work. 

Thank you very much for the time invested in reading the paper and for providing us a list of scientific points to improve the manuscript.

Lines 8. In the abstract, the presentation of the research objectives is not clear. The present formulation does not appear effective.

Following reviewer suggestion, we have rewritten the objective of the abstract among other things. Please see line: 19-37.

Lines 32 and the followings. The introduction does not highlight the background (i.e. context) in which the research is inserted, which must, therefore, be developed into the contents:

• What is the relevance of this paper in regard to the objectives? What is the relevance for the operators of the wine sector and/or research?

• Which are the components of the study’s originality in relation to the knowledge of the aspects highlighted by the research?

• In the final part, it is also pointed out that the structure on which the paper is arranged is not presented.

Thank you for all the points to improve the introduction section. This paper provides understanding of the wine components when interacting with human saliva, and its measurement by instruments and humans. To our best knowledge, although there are scientific works that has studied the relationship of “body” and ethanol glycerol, or polysaccharides, there is not yet any work that integrates them in a model using physiological conditions, as saliva. Saliva, until now, has only been studied in terms of astringency perception. It is believed that further evidence and extensive research on this topic could help wine industry to adjust wine components for a specific mouthfeel characteristic.

Lines 81-156. The methodological part should be reviewed and better structured:

• Material and Methods should be used as a title and Model-wine samples as subtitles (line 81).

Reviewer is right, we apologise for this mistake, it has been corrected.

• There are paragraphs with a few lines and excessive spacing. In the current format, they look quite poor as self-standing sections.

Thank you for noticing.

• It should be both better articulated in the structure and connected to the objectives of the research.

Thank you for the suggestion, we have readjusted for a better match between objectives and materials and methods.

Line 100. In the current format the Table 1 header is not well presented:

• The heading of the titles is not appropriately formatted;

• Mannoproteins lack of measurements indications used;

Reviewer is right, table format was not the adequate and it has been corrected now.

Line 104. Why was a 1/1 dosage chosen in the proportion of saliva addition to wine models? Why have solutions not been prepared with different levels of saliva added to wine models?

• An explanation should be provided to support the choices made in the implementation of the experimental design concerning the objectives of the study.

That is a very interesting point. Until now, the quantity of saliva added in saliva-food study varies, and we could not find any previous study that provide a valid method for quantity of saliva to be added, as it has a huge difference among individuals along the day and depending of the food that one is ingesting. Therefore, we thought that a proportion of 1:1 could give us enough sample to interact fully with the salivary sample.

We feel that to add different levels of saliva to the samples is interesting; especially for populations suffering xerostomia but it is out of the scope of this paper.

Line 150. The defining aspects reported in Table 2 do not appear well conceptualized and should be better specified:

• For example, they recall the term to be defined;

• Concerning the terminology used for the term “body” I do not agree on the definition provided because body is not only defined by the visual aspects, but the taste aspects should also be taken into consideration.

• It is also advisable to add specific references;

We have added further information to the head of Table 2 and into the main test. But we were not completely sure about what the reviewer meant. In this work, the methodology was followed to develop a descriptive sensory profile of the samples. As was mentioned in the text, they were trained by using standards and agree on the attributes and meaning

Lines before 305. A separation between discussions and results can help readers to understand the article.

Thank you for the suggestion, but we feel conservative in the way that the paper is written. As the reviewer can see now, we have modified and increase the discussion of the paper, and with the changes performed we hoped to help the readers

The conclusions should be better linked to the research objectives and the conceptual framework used. We also recommend:

• To better formulate the statement on lines 308-309 which does not appear clear.

• To comment on the difficulties of knowledge in the limitations of the study, through instrumental and sensorial analysis linked to a complex food such as wine, where the interaction between its multiple components is yet the subject of open discussion among researchers.

Thank you for the suggestion; conclusion has been rewritten (see line 438-461)

Even the English language style still needs improvements, to make it more readable. Below one can find some examples:

·         Lines 18 to 20. The sentence has no clear meaning.

·         Lines 75 to 78: Once again the sentence has a very unclear meaning

·         Lines 182 to 189. The words shear rate were repeated 5 times.

Thank you for pointing this out, we have done changes now in these sentences and we have sent it again to an academic English editor.

Reviewer 4 Report

Review:

Oral wine texture perception and its correlation with instrumental texture features of wine-salivamixtures

Overview

This is a rather simple experiment based on the measurement of certain rheological, physical and chemical properties of mixtures of dealcoholized wine with saliva. Even though in the acknowledgements the authors claim one English speaker helped in the edition of the manuscript, this clearly does not show on the manuscript. The manuscript in itself is pervaded by errors, way too many to be listed, and poor English. There is absence of verbs in some phrases, typos in others. The experiment in itself is poorly described and the analysis of the results is simplistic and partial at best. Finally, the findings add nothing to knowledge. The authors have found that tannins correlate with astringency perception, which was the case in the samples containing tannins. This is not novel. As a result, this paper presents more or less preliminary data, that, while interesting, it is just that, preliminary and it should be used as a trampoline for a more complete experimental design.

Specific comments

Numbers signify lines in the text

14-18: two “however” very close to each other, poor English

Abstract: nothing novel reported.

41-48: awkward, long sentence, with poor English. This needs re-writing.

43: what on earth are “wine corks experts”????

62-68: poor English, not clear what the authors mean by “However the polyphenols concentration may influence the perception and so there is a clear difference between white wine and red wine astringency”. I think I know what they mean to say, that is, that red wines are astringency and whites may be not, it is not just what the phrase reads. Have someone speaking and writing English edit this please.

75-80: poor English. Edit and rephrase.

88-91: while I understand that alcohol above 8% may be perceived, this does not preclude the possibility to use higher levels of ethanol, close to real wines. 8% is unrealistic. The authors could have use another alcohol with less aromas. The fact that above 8% is overpowering for the senses does not prevent you to try to run the experiment close to real life conditions (no wines, except certain German Riesling , have 8% ABV).

Table 1: add definition of each acronym at the bottom of the table, just as in figure 2. The same goes for figure 1, add definition of each acronym in figure caption just as in figure 2.

Table 2, in the definition of body, the authors , if they mean in-mouth sensations ,probably may want to say “sloshing” instead of swirling. Swirling is something you do on the glass when you swirl the glass (and not the wine in your mouth as I believe the authors meant to say, which is an action called sloshing). 

Also on table 2, what on earth is the feeling of cereal taste? First, you use the word feeling, alluding to a tactile sensation. But in the definition, you use the word :”taste” which is just that, a taste, but not a tactile sensation. Please clarify.

Figure 1. add acronyms to figure caption just as in figure 2

230-238: the authors need to provide a full mechanistic and chemical explanation for this. this is the short version of it. Ethanol acts as an hydrogen bond disruptor, thereby disrupting or hindering the interaction between salivary proteins, specifically proline rich proteins, and tannins. Mannoproteins act by competing with salivary proteins for tannins and they also decrease the ability of tannins to interact with proteins, thereby modulating astringency. There are several studies that have found what I said and these need to be cited in text.

246-251: the authors failed to consider that they added oak tannins at 1 g/L, which is a lot. As a consequence, some of the flavors, taste and texture of this treatment with oak tannins added may be affected by the inherent flavors of oak tannins. See work from Harbertson and Downey on the effect of exogenous tannin additions on chemistry and flavor (published I believe in Food Chemistry). This is something to be considered, that is, the off flavor contribution of oak derivatives when added to wines.

249: report results in past tense (throughout the text).

Figure 3. add acronyms to figure caption just as in figure 2

Table 5. it is Pearson correlation coefficients. Add “coefficients”.

Literature and references are NOT in the same format. Some journal articles have the first letter capitalized, some others not. Some title articles have the first letter capitalized, some others do not. References should be in the EXACT same format. 

Author Response

Reviewer 4

Oral wine texture perception and its correlation with instrumental texture features of wine-saliva mixtures

Overview

This is a rather simple experiment based on the measurement of certain rheological, physical and chemical properties of mixtures of dealcoholized wine with saliva. Even though in the acknowledgements the authors claim one English speaker helped in the edition of the manuscript, this clearly does not show on the manuscript. The manuscript in itself is pervaded by errors, way too many to be listed, and poor English. There is absence of verbs in some phrases, typos in others. The experiment in itself is poorly described, and the analysis of the results is simplistic and partial at best. Finally, the findings add nothing to knowledge. The authors have found that tannins correlate with astringency perception, which was the case in the samples containing tannins. This is not novel. As a result, this paper presents more or less preliminary data, that, while interesting, it is just that, preliminary and it should be used as a trampoline for a more complete experimental design

We thank the reviewer for the time invested in carefully reading the manuscript and for giving suggestions to improve the paper.

The reviewer is right, the paper has basic methodology and is a first step of a series of studies, but authors believe that the experimental part including instrumental and sensory (trained and expert) is complete enough and gives new information regarding wine mouthfeel.  Additionally, the paper introduces a new concept regarding this relevant topic for wine quality, such as the model of study simulating human physiological conditions. In our opinion, it is the better approach to understand human perception of wine texture.

In order to empathise the effect of other wine components, besides tannins, we have now rewritten entire sections to highlight that new insight into mannoproteins’ role in mouthfeel, and in the study of the relationship between instrumental results (density and rheology) have been done by using two types of panels.

As the reviewer notice, scientific writing for non-native English speakers is one of the biggest struggles, therefore authors contracted a scientific English editor that has reviewed the manuscript once again with the comments of the reviewer 4.

Specific comments

Numbers signify lines in the text

14-18: two “however” very close to each other, poor English

Abstract: nothing novel reported.

Thank you for noticing, authors have looked now for a “however” synonym and change it, we did not realise the importance of connectors variety.

Reviewer is right, the abstract does not reflect the novelty of the work, therefore has been re-writing to reflect the novelty of this work.

41-48: awkward, long sentence, with poor English. This needs re-writing. 

We apologise for this error and have corrected the text as suggested. 

43: what on earth are “wine corks experts”????

In a wine producing country, such as Spain, cork manufacturers play an important role in wineries and wine sensory sessions, and cork quality plays a key role in wine oxygenation.

But considering the reviewer comments, we are aware that in some countries and wine styles, the wine bottles does not have cork, and they have a screw cap. To avoid confusions for international readers, authors have removed the wine cork expert.

62-68: poor English, not clear what the authors mean by “However the polyphenols concentration may influence the perception and so there is a clear difference between white wine and red wine astringency”. I think I know what they mean to say, that is, that red wines are astringency and whites may be not, it is not just what the phrase reads. Have someone speaking and writing English edit this please.

Following the comment of a previous reviewer that section has been already rewritten.  

75-80: poor English. Edit and rephrase.

Thank you for letting us know, we have corrected .

88-91: while I understand that alcohol above 8% may be perceived, this does not preclude the possibility to use higher levels of ethanol, close to real wines. 8% is unrealistic. The authors could have used another alcohol with less aromas. The fact that above 8% is overpowering for the senses does not prevent you to try to run the experiment close to real life conditions (no wines, except certain German Riesling, have 8% ABV).

Fair point, authors did not think to use another alcohol, as this alcohol was already used in their previous work. Initially the ethanol level chosen was 14%, however, that resulted in overpowering the senses due to the flavour intensity of pure ethanol, thus it was not drinkable. For that reason, an ethanol level was chosen on the basis of the minimum alcohol of wine (8%) from a legal perspective (this has been also added to the manuscript)

As authors have this line of research still in progress, for future work, authors will look for other alcohols.

Table 1: add definition of each acronym at the bottom of the table, just as in figure 2. The same goes for figure 1, add definition of each acronym in figure caption just as in figure 2. 

Thank you, it has been added.

 Table 2, in the definition of body, the authors, if they mean in-mouth sensations, probably may want to say “sloshing” instead of swirling. Swirling is something you do on the glass when you swirl the glass (and not the wine in your mouth as I believe the authors meant to say, which is an action called sloshing)

Thank you for the advice, the translation of a Spanish trained panel vocabulary into English is always tricky.

Looking in Cambridge Dictionary authors found that:

Swirl (verb): to move quickly with a twistingcircular movement, or to make something do this.

Slosh (verb) (of a liquid): to move around noisily in the bottom of a container, or to cause liquid to move around in this way by making rough movements.

Therefore, authors were not happy with either of them. Looking in specialise English (USA) wine websites, we have come to the realisation that the word we were looking for was “swishing”, therefore we have corrected in the main manuscript.

Also, on table 2, what on earth is the feeling of cereal taste? First, you use the word feeling, alluding to a tactile sensation. But in the definition, you use the word:” taste” which is just that, a taste, but not a tactile sensation. Please clarify.

Reviewer is right, this paper focus is the mouth feeling (tactile response). But we state that if panellist felt that an attribute, although not tactile, was needed, to mention it, because it was the driver of sample discrimination. Therefore, we encouraged them to describe it, in this case, it was cereal taste, special for those samples containing tannins.

We agree that this can be confusing, and we have added further information (see line 173-177)

Figure 1. add acronyms to figure caption just as in figure 2

Thanks for the suggestion, acronyms have been added

230-238: the authors need to provide a full mechanistic and chemical explanation for this. this is the short version of it. Ethanol acts as a hydrogen bond disruptor, thereby disrupting or hindering the interaction between salivary proteins, specifically proline rich proteins, and tannins. Mannoproteins act by competing with salivary proteins for tannins and they also decrease the ability of tannins to interact with proteins, thereby modulating astringency. There are several studies that have found what I said and these need to be cited in text.

Following reviewer suggestion, that part has been rewritten, more bibliography has been researched and added (please see lines 317 and 348).

246-251: the authors failed to consider that they added oak tannins at 1 g/L, which is a lot. As a consequence, some of the flavours, taste and texture of this treatment with oak tannins added may be affected by the inherent flavours of oak tannins. See work from Harbertson and Downey on the effect of exogenous tannin additions on chemistry and flavour (published I believe in Food Chemistry). This is something to be considered, that is, the off-flavour contribution of oak derivatives when added to wines.

Reviewer is right, tannin added to create a model wine were right, but considering that in red wines the ranges oscillated from 1600 to 2500 mg/L we thought that this quantity was sensible

Thank you for the suggestion. We did not read before this paper, and it has meaningful information, however we did not see how can provide further information as this paper already deals with model-wines and all the ingredients are “artificially added”. We have added though in the conclusion the need to do the study with real wines.

249: report results in past tense (throughout the text).

In this version, we have reviewed the tense of the manuscript.

Figure 3. add acronyms to figure caption just as in figure 2

Thanks for the suggestion, acronyms have been added

Table 5. it is Pearson correlation coefficients. Add “coefficients”.

Thanks for the suggestion, it has been added

Literature and references are NOT in the same format. Some journal articles have the first letter capitalized, some others not. Some title articles have the first letter capitalized, some others do not. References should be in the EXACT same format. 

Although we have used EndNote software, we have reviewed and update the references, thank you for noticing this small detail to make our paper more consistent.

Round  2

Reviewer 3 Report

At the second review, the paper appears to have been sufficiently improved.

Kind Regards